# π–π Stacking Interaction of Metal Phenoxyl Radical Complexes

**DOI:** 10.3390/molecules27031135

**Published:** 2022-02-08

**Authors:** Hiromi Oshita, Yuichi Shimazaki

**Affiliations:** 1Center for Integrative Quantum Beam Science (CIQuS), Institute of Materials Structure Science (IMSS), High Energy Accelerator Research Organization (KEK), 1-1 Oho, Tsukuba 305-0801, Ibaraki, Japan; hiromi.oshita@kek.jp; 2Graduate School of Science and Engineering, Ibaraki University, Bunkyo, Mito 310-8512, Ibaraki, Japan

**Keywords:** π–π stacking interaction, phenoxyl radical, indole, metal complex, oxidation

## Abstract

π–π stacking interaction is well-known to be one of the weak interactions. Its importance in the stabilization of protein structures and functionalization has been reported for various systems. We have focused on a single copper oxidase, galactose oxidase, which has the π–π stacking interaction of the alkylthio-substituted phenoxyl radical with the indole ring of the proximal tryptophan residue and catalyzes primary alcohol oxidation to give the corresponding aldehyde. This stacking interaction has been considered to stabilize the alkylthio-phenoxyl radical, but further details of the interaction are still unclear. In this review, we discuss the effect of the π–π stacking interaction of the alkylthio-substituted phenoxyl radical with an indole ring.

## 1. Introduction

Weak interactions in biological system are important in terms of the structural and functional aspects [1,2,3,4,5,6,7,8,9,10,11,12,13,14,15,16,17,18,19,20,21,22,23,24,25,26,27,28]. Of particular significance are hydrogen bonds [19,20], π–π stacking interactions [12,13,14] and other non-covalent interactions, including cation-π, NH-π and CH-π interactions [21,22,23,24,25,26,27,28]. These interactions greatly contribute to the construction and stabilization of the highly ordered structures of proteins and other biological molecules, molecular recognition of the substrate and catalytic activity of enzymes [1,2,3,4,5,6,7,8,9,10,11,12,13,14,15,16,17,18,19,20,21,22,23,24,25,26,27,28]. From the importance of weak interactions, the functionalization of the metal complexes has been investigated. In particular, the hydrogen bond was found to stabilize the active dioxygen species [29,30,31]. Thus, the metal complexes with some groups capable of hydrogen bond formation exhibited novel properties and reactivities. On the other hand, the other weak interactions, such as π–π stacking interaction, have remained yet to be employed for the functionalization of the artificial metal complexes.

In order to find the way to the functionalization of metal complexes by π–π stacking interaction, we focused on model studies of the single copper enzyme, galactose oxidase (GO), which catalyzes primary alcohol oxidation to give the corresponding aldehyde [32,33,34,35,36,37]. The catalytic alcohol oxidation mechanism involves the hydrogen atom abstraction from the π-position of primary alcohol to the phenoxyl radical bound to the copper(II) ion in the active form of GO [35]. The active site structure of GO has been revealed to have a square pyramidal structure with two imidazole moieties of the histidine residues and two phenol moieties of the tyrosine residues coordinated to the copper(II) ion (Figure 1) [36]. One of the two phenol moieties is deprotonated and bound at an equatorial position of the copper center as a phenolate ligand, to which the sulfur atom derived from the cysteine residue is bound at the *ortho*-position to form an *ortho*-alkylthiophenolate moiety. The active form of GO is known to be the copper(II)-alkylthiophenoxyl radical, and this phenoxyl radical has been proposed to be stabilized by π–π stacking interaction with the indole ring of the proximal tryptophan residue (Trp 290) [37].

A number of model studies of GO have been reported with emphasis on the characteristic electronic structure of copper(II)-phenoxyl radical and its reactivity [37,38,39,40,41,42,43,44,45,46,47]. Copper(II)-phenoxyl radical complexes have quite different geometric and electronic structures compared with the copper(III)-phenolate species, which is the same one-electron oxidized form of the copper(II)-phenolate species [48,49,50,51,52,53]. The difference is reflected to the reactivity on the primary alcohol oxidation; the copper(III)-phenolate is less reactive for the primary alcohol oxidation in comparison with the copper(II)-phenoxyl radical, suggesting that the hydrogen abstraction by the phenoxyl radical is significant in the course of the oxidation [52,53,54]. However, the effect of the π–π stacking interaction of the phenoxyl radical with the indole ring is still unclear.

With these points in mind, we focus on the effects of the π–π stacking interaction of the phenoxyl radical bound to a copper ion. The geometric and electronic structural changes of the phenoxyl radical by the π–π stacking interaction and its effects on reactivity are discussed on the basis of recent results. This review discusses π–π stacking interactions of the phenoxyl radical species with the indole ring around the metal center.

## 2. The π–π Stacking Interaction in the Active Site of GO

One of the aromatic amino acids tryptophan (Trp) has a side chain indole ring, which is a fused-ring aromatic molecule composed of a pyrrole and a benzene ring. The indole ring has been considered to stabilize the active species or to be involved in electron transfer pathways by π–π stacking interaction with the other aromatic ring. Therefore, it is frequently located at the proximal position of the active or important site of enzymes [41,55]. GO catalyzes oxidation of the primary alcohol at the 6-position of galactose to generate the corresponding aldehyde [35,56], and the active species of the catalytic oxidation has been well known to be the copper(II)-phenoxyl radical derived from the one-electron oxidation of the phenolate moiety of tyrosine 272 (Tyr 272) bound to the copper(II) ion [32,33,34,35,36,37]. The phenoxyl radical moiety of Tyr 272 is modified by the carbon-sulfur covalent bond at the *ortho*-position, derived from the cross linking with the cysteine 228 (Cys 228), and the active form of GO is described as a copper(II)-alkylthiophenoxyl radical species [51]. The indole ring of Trp 290 is located in the second coordination sphere of the active site of GO, most probably for stabilization of the copper(II)-alkylthiophenoxyl radical species by the π–π stacking interaction with the alkylthiophenoxyl radical of Tyr 272 [57,58,59,60,61]. Previous studies using the GO mutants with Trp 290 replaced with various other amino acids, such as glycine (gly), phenylalanine (Phe) and histidine (His), revealed a shorter lifetime of the Cu(II)-alkylthiophenoxyl radical, indicating that the π–π stacking interaction plays an important role in the stabilization of the radical in the active form of GO [58,59,60].

Apo-GO, which has no Cu(II) ion in the active site, shows a significantly different structure, especially in the active site, where the phenoxyl radical could not be detected and no C-S bond formation in the phenol moiety of Tyr 272 was found (Figure 1) [60,61]. In addition, the π–π stacking interaction of the indole moiety of Trp 290 with the phenol moiety of Tyr 272 was not observed, and the indole ring was exposed to the outer sphere of protein [60]. However, both the stacking interaction and the C-S cross link were generated upon the addition of copper(II) ion to the apo-GO by the soaking experiments [60]. These results suggests that the indole ring of Trp 290 selectively interacts with the alkylthio-phenoxyl radical moiety.

On the other hand, the effect of the C-S cross link of the alkylthiophenoxyl radical have been also reported [62]. The mutation study replacing Cys with glycine (Gly) and serine (Ser) of the GO homologue, GlxA, revealed that the unpaired electron of the phenoxyl radical was transferred to the indole moiety of Trp to form the indolyl or indole–π–cation radical [62]. The result suggests that the alkylthiophenoxyl radical is also stabilized by the C-S cross link. It was proposed that in the absence of the C-S cross link, the indole ring of Trp may be also located in the proximal position of the phenoxyl radical, but that the conformation of the indole moiety is different from that of the native GO. The indole ring was found to be in a position not suitable for the π–π stacking interaction, and thus it was rotated to form the pseudo-physical (van der Waals) interaction mainly of the indole C–H bonds with the phenolate moiety (Figure 2). The experimental and calculation studies of the mutants proposed that the spin density in these mutants is distributed mainly on the indole moiety coupled with the Tyr phenolate moiety [62]. Thus, the methylthio group is important for the stabilization of the phenoxyl radical by the face-to-face π–π stacking interaction, and, for easy transfer of the unpaired electron of the phenoxyl radical, may be easily transferred to the indole ring by rotation.

## 3. π–π Stacking Interaction of Methylthio-Phenoxyl Radical Metal Complexes

The π–π stacking interaction between two phenoxyl radicals is slightly different from that involving the neutral phenol and phenolate anion, due to the existence of one unpaired electron on the π conjugated system. The π orbital is assigned to the SOMO on the phenoxyl radical as whole; therefore, a SOMO–SOMO interaction can be considered in general [63]. Such a SOMO–SOMO interaction could not be observed in the crystals of metal phenoxyl radical complexes except alkylthio-phenoxyl radical species [39,40,64,65,66]. In many cases, the phenoxyl radical moiety interacted only with the counter anion weakly [39,40,64]. On the other hand, the alkylthio-phenoxyl radical, which is similar to the phenoxyl radical in the active form of GO, showed a significant interaction with the other phenoxyl radical by π–π stacking interaction in the solid state [65]. Furthermore, the stacking interaction between alkylthio-phenoxyl radicals exhibited a different dependence on the central metal ion [65,66].

Recently, the X-ray crystal structures of oxidized copper(II)- and nickel(II)-diphenolate complexes have been successfully determined. One-electron oxidized copper(II) di(methylthiophenolate) complex with a salen-type ligand, [Cu(MeS-salen)]SbCl_6_ (**[1]SbCl_6_**), was prepared by reaction of Cu(MeS-salen) (**1**) with one equivalent of thiantrenyl radical hexachloroantimante salt (Th^+^SbCl_6_^−^) (Figure 3) [65]. Similar oxidation by addition of one equivalent of Th^+^SbCl_6_^−^ to the solution of Ni(MeS-salen) (**2**) afforded the one-electron oxidized complex, [Ni(MeS-salen)]SbCl_6_ (**[2]SbCl_6_**) [66]. The Cu and Ni K-edge X-ray absorption near edge structures (XANES) of these one-electron oxidized complexes showed no significant difference from that of the complexes before oxidation, whereas the sulfur K-edge XANES exhibited a noticeable increase in the pre-edge intensity of **[1]SbCl_6_** [65,66,67,68]. These results indicate that the one-electron oxidized complexes **[1]SbCl_6_** and **[2]SbCl_6_
** are ligand centered oxidation species and thus can be assigned to the metal(II)-phenoxyl radical complexes.

The X-ray crystal structure of **[1]SbCl_6_** revealed that it has half of the molecule as a crystallographically independent unit, indicating that **[1]SbCl_6_** has a perfect *C*_2_ axis through the center of the ethylenediamine and the copper ion (Figure 2A) [65]. The results indicate that the structural features of the two phenolate moieties are identical. The bond length of Cu–O in **[1]SbCl_6_** (1.920 (1) Å) was longer than that of **1** (1.902(1) Å), and that of C–O of the phenolate moiety in **[1]SbCl_6_
**(1.295(2) Å) was shortened compared with **1** (1.307(2) Å). On the other hand, the X-ray crystal structure of **[2]SbCl_6_** showed no symmetry axis, indicating that the two phenolate moieties in **[2]SbCl_6_** are structurally not identical (Figure 2B) [66]. The differences of the two Ni–O and two C–O bond lengths in **[2]SbCl_6_** (Ni–O(1) 1.875(2) Å, Ni–O(2) 1.843(2) Å; C(1)–O(1) 1.279(4) Å, C(2)–O(2) 1.315(4) Å) were larger than the differences observed between **1** and **[1]SbCl_6_**. The bond lengths of Ni–O(2) and C(2)–O(2) in **[2]SbCl_6_** were similar to those of complex **2** (Ni–O 1.8586(9) Å, C–O 1.312(1) Å), supporting that one of the phenolate moieties in **[2]SbCl_6_** maintains the phenolate electronic structure, so that **[2]SbCl_6_** can be described as the nickel(II) localized phenoxyl radical complex, [Ni^II^(phenolate)(phenoxyl radical)]^+^ [66]. In the case of copper complex **[1]SbCl_6_**, the difference of the Cu–O and C–O bond lengths was rather small in comparison with the nickel complexes. From the results together with the symmetric features in crystals, **[1]SbCl_6_** can be assigned to the copper(II)-delocalized radical species described as [Cu^II^(0.5-phenoxyl radical)_2_]^+^ with the electron equally distributed on the two phenolate moieties [65].

The crystal packing of these one-electron oxidized complexes revealed the existence of the intermolecular interaction. The crystal packing of **[1]SbCl_6_** showed the intermolecular π–π stacking interaction between the two half-phenoxyl radical moieties of the neighboring molecules with the distance of 3.18 Å, resulting in the one-dimensional chain formation. Furthermore, the sulfur atom of the 0.5-alkylthiophenoxyl radical was in close contact with the central copper ion with the distance of 3.04 Å (Figure 3A,B) [65]. On the other hand, the crystal packing of the localized phenoxyl radical complex **[2]SbCl_6_** showed a different view of the crystal packing (Figure 3C,D). The intermolecular π–π stacking interaction was observed between the localized phenoxyl radical moieties of the neighboring molecules with the distance of 3.1 Å to give a dimerization species, but no one-dimensional chain formation could be detected [66].

Such a difference in the intermolecular π–π stacking interaction of copper and nickel complexes was considered to arise from the difference in the population of the unpaired electron spin on the phenoxyl radical moiety. In general, the SOMO–SOMO interaction shows some characteristics [63]; π–π stacking geometry involving the SOMO–SOMO interaction exhibits the atom-over-atom configurations with a distance shorter than the sum of the van der Waals radii (less than 3.19 Å in the case of the C–C distance), as opposed to the atom-over-bond or atom-over-ring configurations, which are typical of van der Waals π–π stacking [14,63,69]. Furthermore, possible minor deviations from the planarity of the constituent molecules indicate the primary role of the SOMO–SOMO interaction [63]. In the case of **[2]SbCl_6_**, the intermolecular π–π stacking interaction between two localized phenoxyl radical moieties can be assigned to the characteristic SOMO–SOMO interaction, the atom-over-atom configurations being observed with the shortest C–C distance of 3.1 Å and small deviation of the stacked phenoxyl radical moieties from planarity (Figure 3C,D) [66]. On the other hand, **[1]SbCl_6_** having two half-phenoxyl radical moieties showed the intermolecular interaction typical of van der Waals π–π stacking, which is very similar to that before oxidation (Figure 4) [65]. These results suggest that the alkylthio-phenoxyl radical moiety prefers the π–π stacking interaction and that the mode of the stacking depends on the electronic structure of the phenoxyl radical moiety.

The π–π stacking interaction can give rise to a different electronic structure of the metal phenoxyl radical species. The phenoxyl radical localization and delocalization can be assigned by the NIR band intensity and band width, which is similar to the mixed valence dinuclear complexes [70,71,72]. In the case of the completely localized phenoxyl radical unpaired electron on one of the two phenolate moieties, no characteristic band is observed in the NIR region. However, an increase in the degree of the delocalization gradually causes the appearance of the NIR band, and the full delocalization of the radical unpaired electron on the two phenolate moieties described as (0.5-phenoxyl radical)_2_ gives a sharp intense band with a narrow bandwidth in the NIR region [73,74,75]. The UV-vis-NIR absorption spectrum of the CH_2_Cl_2_ solution of **[1]SbCl_6_** showed a broad NIR band with a small intensity at 9100 cm^−1^, which is a feature similar to the one-electron oxidized methoxy-substituted diphenolate copper(II) complex, [Cu(MeO-salen)]^+^ (Figure 5A) [65,73]. The band was assigned to the phenolate to phenoxyl radical charge transfer, which was also supported by TD-DFT calculation. On the other hand, the solid sample of **[1]SbCl_6_** showed a large shift of the NIR band to 5000 cm^−1^, which is different from that of the solid sample of [Cu(MeO-salen)]^+^ (7800cm^−1^ in CH_2_Cl_2_ vs. 7200 cm^−1^ in the solid state) [65,73]. Furthermore, the bandwidth of the NIR band of **[1]SbCl_6_** in the solid state was narrow in comparison with the solid sample of [Cu(MeO-salen)]^+^ and the CH_2_Cl_2_ solution of **[1]SbCl_6_** (Figure 5B) [65,73]. These results indicate that **[1]SbCl_6_** has different electronic structures, the localized phenoxyl radical in CH_2_Cl_2_ and the delocalized phenoxyl radical on two phenolate moieties in the solid state due to the stacking interaction, which was supported by DFT calculations [65].

The difference arising from the stacking interaction was also observed in EPR measurement. In general, copper(II)-phenoxyl radical complexes are EPR inactive or show the characteristic EPR signal at ca. *g* = 4, due to the magnetic coupling of two unpaired electron spins between the copper d-electron and the phenoxyl radical electron, resulting in *S* = 0 or 1 in total [39,76,77]. However, the solid sample and the frozen sample of the CH_2_Cl_2_ solution of **[1]SbCl_6_** showed a significant isotropic EPR signal at ca. *g* = 2.0, which is of similar intensity to the signal of the di(phenoxyl radical) copper(II) complex of the same ligand, **[1](SbCl_6_)_2_** (Figure 6) [65,67,68]. These results can be considered to show that all unpaired electron spins in both the copper(II) ion and phenoxl radical line up ferromagnetically in the one-dimensional chain of **[1]SbCl_6_** by the π–π stacking interaction, which is similar to the other ferromagnetic one-dimensional chain compounds. In fact, the solid sample of **[1]SbCl_6_** exhibited the ferromagnetic interaction intra- and inter-molecularly [65]. Thus, the π–π stacking interaction of the methylthio-phenoxyl radical moiety in **[1]SbCl_6_** influences the electronic structure.

## 4. The π–π Stacking Interaction of Phenoxyl Radical with an Indole Ring

In order to understand π-π stacking properties of the indole ring with the phenoxyl radical, copper(II) and nickel(II) diphenolate salen-type complexes with two *para*-methoxy- or *para*-methylthio-phenolate moieties and a side chain indole ring on the ethylenediamine backbone were synthesized and characterized (Figure 7) [78,79,80]. Their one-electron oxidized complexes were characterized, particularly focusing on the different behavior of the indole ring due to the oxidation state.

### 4.1. The π–π Stacking Interaction of Methoxyphenoxyl Radical with an Indole Ring

The X-ray crystal structure of copper(II) salen-type complex **3** having a pendent indole ring and methoxy substitution at *para*-position of two phenolate moieties revealed no significant intra- and inter-molecular interaction (Figure 8A), and thus the coordination sphere of complex **3** was very similar to that of the MeO-salen complex without the pendent indole ring Cu(MeO-salen) [65,73]. This result well supports the previous reports that the indole ring does not interact with an electron-rich aromatic ring, such as the phenolate moiety [79,80,81,82,83,84,85]. Complex **3** was oxidized by the addition of one equivalent of AgSbF_6_ as a one-electron oxidant in CH_2_Cl_2_ to form the one-electron oxidized complex **[3]SbF_6_**, whose X-ray crystal structure revealed that **[3]SbF_6_** consists of two crystallographically independent species in the unit cell (Figure 8B,C) [78]. They showed similar structural characteristics in the first coordination sphere, and one of the two Cu-O bonds in each species was ca. 0.1 Å longer than the other, indicating that the radical electron was fully localized on one of the phenolate moieties. Therefore, the two complexes in the unit cell could be assigned to the localized phenoxyl radical complexes described as [Cu^II^(phenolate)(phenoxyl radical)]^+^ [64]. The position of the indole ring in **[3]SbF_6_** was completely different from that in the neutral complex **3**; the indole ring was found to lie on the phenoxyl radical moiety of salen ligand by the π–π stacking interaction in both complexes. The distance between indole ring and the phenoxyl radical moiety was determined to be ca. 3.4 Å, which is in the acceptable range of the π–π stacking interaction [14,86]. Furthermore, this stacking conformation of the indole ring and phenoxyl radical moiety showed the atom-over-bond or atom-over-ring configuration, which is different from the π–π stacking interaction by SOMO–SOMO overlapping [63]. These results indicate that the indole ring recognizes the phenoxyl radical moiety and forms a π–π stacking structure with it selectively.

The oxidized complex **[3]^+^** also showed a different behavior in CH_2_Cl_2_ in comparison with the complex having a side chain methyl group, [4]**^+^ [78]**. The CH_2_Cl_2_ solutions of **[3]^+^** and **[4]^+^** exhibited the specific band in the NIR region, which was assigned to the phenolate to phenoxyl radical intervalence charge transfer indicating the relatively localized Cu(II)-methoxyphenoxyl radical on one of the phenolate moieties (Figure 9A) [46,53,66,67,68]. Notably, the NIR band of **[3]^+^** (8700 cm^−1^) shifted to the higher energy region by 1100 cm^−1^ compared with that of **[4]^+^** (7600 cm^−1^), showing that the phenoxyl radical moiety was affected by the pendent indole ring. The energy difference could be estimated from this NIR band difference to be 13 kJ/mol, which is in line with the energy of the π–π stacking interaction (4–20 kJ/mol) [14,78,86]. Therefore, the indole-phenoxyl radical stacking interaction was also maintained in the CH_2_Cl_2_ solution. It has been reported that the π–π stacking interaction of the indole ring depends on the solvent properties [12,87]. In CH_3_CN (dielectronic parameter [1/η_2_ − 1/ε] of CH_3_CN: 0.528; CH_2_Cl_2_: 0.382) a similar NIR band shift to the higher energy region was also observed for **[3]^+^** (9500 cm^−1^) in comparison with **[4]^+^** (8700 cm^−1^) [78]. In toluene (dielectric parameter [1/η_2_ − 1/ε]: 0.026), however, the NIR band peak difference between **[3]^+^** (8060 cm^−1^) and [Cu(MeO-salen)]^+^ (7880 cm^−1^) [65,73,88] was rather small (Figure 9B), and the energy was estimated to be 2.4 kJ/mol. These results are consistent with the fact that the π–π stacking interaction is more favored in the polar solvent [78]. Furthermore, toluene, being an aromatic solvent, may be also effective for inhibition of the intramolecular π–π stacking interaction by solvation.

DFT calculations supported the intramolecular π–π stacking interaction of the indole ring with the phenoxyl radical. The calculation results suggested that the π–π stacking form is more stable than the open structure having no significant stacking interaction [78]. Furthermore, TD-DFT calculation of **[3]^+^** showed that the NIR band of **[3]^+^** consists of the two characteristic components at 7175 cm^−1^ and 9291 cm^−1^ in CH_2_Cl_2_. The 7175-cm^−1^ band could be assigned to the intervalence charge transfer (IVCT) band from the phenolate to the phenoxyl radical moiety as the transition from HOMO to LUMO [78]. The assignment is in good agreement with the NIR peak of the complexes without a pendent indole moiety, **[4]^+^** and [Cu(MeO-salen)]^+^ [65,73]. On the other hand, the 9291-cm^−1^ band was assigned to the charge transition from the indole moiety to the phenoxyl radical moiety. Such a charge transfer band was also suggested at 28428 cm^−1^ (352 nm) by TD-DFT calculation (Figure 10), which agrees well with the previous report on the appearance of the charge transfer band in the near UV region due to aromatic ring stacking [12,79,87]. In fact, the different UV-vis spectrum between **[3]^+^** and **[4]^+^** exhibited a band at 352 nm in CH_2_Cl_2_, and the CD spectrum of **[3]^+^** showed a significant CD peak at ca. 350 nm, which is considered to be due to the proximal effect on the charge transfer band. The difference UV-vis spectrum and the CD peak were solvent-dependent, and the UV band shift and the CD peak could not be detected in toluene [78]. From the experimental and calculation results, the indole ring was concluded to stabilize the phenoxyl radical by the π–π stacking interaction, where it serves as an electron-donor to the phenoxyl radical moiety [78].

### 4.2. The π–π Stacking Interaction of Methylthiophenoxyl Radical with an Indole Ring

The methylthiophenoxyl radical complex **[1]SbCl_6_** favors the π–π stacking interaction in the solid state. As discussed in the previous section, its electronic structure is changed from the localized phenoxyl radical species to the delocalized species by the stacking interaction [65]. The effect of the proximal indole ring in the methylthiophenoxyl radical complex could be clarified by characterization of complex **5** having the same coordination structure as **1** but with a pendent indole ring.

One-electron oxidized complex **[5]^+^** could be generated by the addition of 1 equivalent of thianthrenyl cation radical salt (Th^+^SbCl_6_^−^) to the CH_2_Cl_2_ solution of **5**. The UV-vis-NIR spectrum of **5** showed a spectral feature similar to that of complex **[1]SbCl_6_** in CH_2_Cl_2_, suggesting that the phenoxyl radical was localized on one of the phenolate moieties in CH_2_Cl_2_ [65]. In comparison with the other phenoxyl radical complexes having a side chain, such as a methyl (**[6]^+^**) and a phenyl (**[7]^+^**) group, complex **[5]^+^** exhibited the NIR band assigned to the phenolate to phenoxyl radical IVCT at a quite different position [79]. The NIR band of complex **[5]^+^** shifted to the higher energy region than the bands of **[6]^+^** and **[7]^+^** by ca. 1000 cm^−1^ (9.6 kJ/mol). This energy difference is in line with the energy of π–π stacking interaction (4–20 kJ/mol) and the stabilization energy of the methoxy-substituted complex **[3]^+^**, indicating that the pendent indole ring of **[5]^+^** contacts with the phenoxyl radical moiety by the π–π stacking interaction in CH_2_Cl_2_ solution [14,78,79,86]. Considering the difficulty of the stacking interaction between the two methylthiophenoxyl radical complexes in CH_2_Cl_2_ solution of **[1]SbCl_6_**, the interaction of the indole ring is more effective for perturbation of the methylthiophenoxyl radical [65,79]. On the other hand, it is noticed that the phenyl group in **[7]^+^** may not be effective for the perturbation by the stacking interaction [79].

The methylthiophenoxyl radical complex **[1]^+^** showed a significantly intense EPR signal at ca. *g* = 2 in frozen solution, though copper(II)-phenoxyl radical complexes such as **[1]^+^** having a two-spin system with the magnetic interaction are generally EPR-silent [65,67,68]. This can be considered to be due to the formation of a polymerization structure in the frozen solution by intermolecular π–π stacking interaction, and thus the intense EPR signal of the frozen sample of **[1]^+^** may arise from various magnetic interactions in the polymerization structure [65]. Although complex **[6]^+^** showed a similar intensity of the EPR signal at *g* = ca. 2, complexes **[5]^+^** and **[7]^+^** exhibited the EPR intensity decrease estimated to be more than 70% in the case of **[5]^+^** [79]. The result indicates that the stacked indole ring and phenyl group inhibited the formation of the polymerization structure by the intermolecular stacking between the methylthiophenoxyl radical moieties, and as a result, the electronic structure change of the methylthiophenoxyl radical to form the localized phenoxyl radical species on one of the phenolate moieties was also inhibited [79]. DFT calculation using gradient isosurfaces methods[89] suggested that complexes **[5]^+^** and **[7]^+^** could form the π–π stacking structure of the methylthiophenoxyl radical with the pendent aromatic ring, with the indole ring showing a larger degree of the interaction than the phenyl ring (Figure 11) [79]. Thus, the π–π stacking interaction of the pendent indole moiety with the methylthiophenoxyl radical is more effective than the intermolecular stacking between two methylthiophenoxyl radical moieties.

### 4.3. The Effect of the Electronic Structure of the Phenoxyl Radical on π–π Stacking Interaction with an Indole Ring

The above-mentioned discussions in this review focused mainly on the π–π stacking interaction of the localized phenoxyl radical copper(II) complexes [78,79]. The ligand centered oxidation metal complexes with the salen-type ligand sometime show that the radical unpaired electron spin is delocalized on the two phenolate moieties [46,64,66,85,90,91,92,93]. The phenoxyl radical localization and delocalization depend on the central metal ion. Copper(II) complexes generally favor the localized phenoxyl radical described as [Cu^II^(phenolate)(phenoxyl radical)]^+^ [46,53,66,67,68], whereas some nickel complexes show the delocalization of the radical to form [Ni^II^(0.5-phenoxyl radical)_2_]^+^ configuration [46,64,66,75,81,93,94]. In this section, the effect of the π–π stacking interaction of the indole ring with the 0.5-phenoxyl radical moiety is discussed for the Ni(II) complex of the ligand of complex **3** having a pendent indole ring as an example [80].

The neutral and the methoxyphenoxyl radical nickel(II) complexes, **8** and **[8]SbF_6_**, were prepared by the procedures similar to those employed for the isolation of copper(II) complexes **3** and **[3]^+^**, respectively, in 1,1,2,2-tetrachloroethane(C_2_H_2_Cl_4_)/*n*-hexane. The X-ray crystal structures of complexes **8** and **[8]^+^** revealed that the structures are very similar to those of complexes **3** and **[3]SbF_6_**, respectively (Figure 12) [78,80]. Although the neutral complex **8** showed no significant interaction of the pendent indole ring intra- and intermolecularly, the crystal structure of **[8]SbF_6_** consisted of two different stacking structures in the unit cell as revealed for **[3]SbF_6_**, the position of the phenolate ring stacked with the indole moiety being different. In addition, details of the coordination structures of the two molecules in the unit cell are different. The two Ni–O lengths of **[8]SbF_6_** (a) were the same (1.841(9) Å), whereas they were slightly different in the other molecule **[8]SbF_6_** (b) (1.833(7) Å and 1.860(7) Å). The bond length between copper ion and the phenoxyl radical oxygen atom is ca. 0.1 Å longer than that between copper ion and the phenolate oxygen atom in complex **[3]^+^**, which was assigned to the phenoxyl radical localized on one of the phenolate moieties [80]. Form the results, complex **[8]SbF_6_** was mainly assigned to the radical with the unpaired electron spin delocalized on two phenolate moieties, described as [Ni(0.5-phenoxyl radical)_2_]. On the other hand, complex **[8]^+^**, which has the structure showing no significant interaction of the pendent indole ring with the phenoxyl radical moiety, could be isolated by changing the solvent from 1,1,2,2-tetrachloroethane to chloroform. (Figure 12B). This result suggests that the π–π interaction in this complex may be less favored than that in the localized phenoxyl radical complex [3]**^+^**[80].

Delocalization of the unpaired electron on the two phenolate moieties in **[8]SbF_6_** was maintained in CH_2_Cl_2_ solution, where **[8]SbF_6_** showed the intense NIR band at 4700 cm^−1^ (ε = 16800 M^−1^cm^−1^) [78,79]. The intensity of this band was much stronger than that of the localized phenoxyl radical copper(II) complexes (Figure 13) [80]. The NIR band feature with a strong intensity and a narrow bandwidth could be assigned to the fully delocalized system based on the mixed valence systems by Robin and Day classification [46,53,66,67,68,69,70,71,72]. Therefore, the radical unpaired electron fully delocalized on two phenolate moieties and described as [Ni(0.5-phenoxyl radical)_2_]^+^ is maintained in the CH_2_Cl_2_ solution, which is in good agreement with the other Ni^II^-phenoxyl radical species of the salen-type ligands.

On the other hand, the indole ring in the phenoxyl radical complex **[8]^+^** may be considered to be in close contact with the 0.5-phenoxyl radical moiety in the CH_2_Cl_2_ solution, whereas the experimental difference between complexes with and without a pendent indole ring was rather small. The NIR band in the UV-vis-NIR spectrum of Ni^II^-phenoxyl radical of [8]**^+^**exhibited a slight decrease in the intensity at 4700 cm^−1^ compared with that of **[9]^+^** but a slight intensity increase at 10200 cm^−1^ (Figure 13B) [80]. The results indicated that the effect of the π–π stacking interaction of the indole ring with the phenoxyl radical in complex **[8]^+^** was rather small as compared with that in the copper complex **[3]^+^** [78,80]. In fact, the complex **[8]^+^**with an open structure could be isolated by using a different solvent. Thus, the π–π stacking interaction of the indole ring with the 0.5-phenoxyl radical moiety is less effective than that with the localized phenoxyl radical in the symmetric two-phenolate ligand system [80].

TD-DFT calculation of complex **[8]^+^** suggested that the NIR band characteristics of **[8]^+^** are different from those of the copper(II)-phenoxyl radical complexes and the oxidized nickel(II)-salen complexes without the pendent indole moiety. Especially, the NIR band of **[8]^+^** at 10686.0 cm^−1^ was predicted as a characteristic transition from *β*HOMO-4 to LUMO. The contribution of the Ni ion orbital in *β*HOMO-4 was estimated to be ca. 47%, whereas the Ni ion contribution in LUMO was only 13% (Figure 14) [80]. Therefore, this band could be described as a MLCT band from the Ni(II) ion to the indole ring. It should be mentioned in this connection that the electronic structure of the one-electron oxidized Ni-salen complexes could change from the Ni(II)-phenoxyl radical to the Ni(III)-phenolate state by addition of exogenous ligands [94,95]. In addition, the transition at 9693.1 cm^−1^ was predicted to be the LLCT band, which could be described as the charge transfer from the indole to the delocalized phenoxyl radical (Figure 14) [80]. These CT-bands characteristics at ca. 10000 cm^−1^ support the close contact of the indole ring with the coordination plane in **[8]^+^**. Thus, theoretical calculations showed that the indole moiety selectively interacts with the phenoxyl radical moiety and stabilizes the nickel(II)-phenoxyl radical complex by the π–π stacking interaction.

## 5. Summary and Conclusions

The roles of the π–π stacking interaction in the metal-phenoxyl radical complexes have been discussed in this review, especially focusing on the interaction of the alkylthiophenoxyl radical with the indole ring as seen in the single copper enzyme GO. GO has an alkyltiophenoxyl radical bound to copper ions in the active site, and the indole moiety of Trp 290 located at the proximal position of the phenoxyl radical is involved in the π–π stacking interaction. The alkylthio group is important for the stabilization of the phenoxyl radical state. In the absence of the alkylthio group, the phenoxyl radical is less stable and is reduced by the electron transfer from indole, where the indole ring showed a different mode of interaction from the π–π stacking interaction in the native form of GO.

The alkylthio-phenoxyl radical can be stabilized by the π–π stacking interaction, which gives rise to a slightly different electronic structure. However, the electronic structure and the effect of the π–π stacking interaction depend on the central metal ion. In contrast with the van der Waals π–π stacking interaction in the copper complexes, the nickel complex exhibits the SOMO–SOMO interaction. The π–π stacking interaction of the phenoxyl radical with the indole ring significantly stabilizes the phenoxyl radical state, and the indole to phenoxyl radical charge transfer can be detected in the NIR region in the absorption spectrum. This assignment may be taken to show the close energy gap between indole and phenoxyl radical. Therefore, a small perturbation of the phenoxyl radical can lead to the electron transfer. These observations indicate that the π–π stacking interaction of phenoxyl radical with the indole ring is more effective than that between two phenoxyl radicals, and that the electronic structures of the phenoxyl radical can be controlled by the π–π stacking interaction with the indole moiety.

Taken together, the π–π stacking interaction of the phenoxyl radical with the indole ring is important in the active site of GO, and we believe that unique properties of the π–π stacking interaction involving phenoxyl, indole and various other aromatic rings may lead to novel functionalization of the metal complexes.

## Data Availability

Data is contained within the article.

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
