# Peer review of "π–π Stacking Interaction of Metal Phenoxyl Radical Complexes"

_molecules, 2022, doi:10.3390/molecules27031135_

Round 1

Reviewer 1 Report

The article by Oshita and co-workers shows an interesting review considering recent studies on phenoxyl radical complexes as galactose oxidase models and the authors have performed a complete discussion of the effect of pi-stacking interactions between the phenoxyl radicals and the indole ring. the article is well-written and presented and I recommend its publication in Molecules journal in the present form.

Author Response

Thank you very much for your positive comments.

Reviewer 2 Report

Role of pi-pi stacking interaction in metal phenoxyl radical complexes is well discussed. This review can be published in Molecules.

Books on noncovalent interactions (including cation-pi interactions)

[1]   P. Hobza and K. Müller-Dethlefs, Non-covalent Interactions: Theory and Experiment, Royal Society of Chemistry, 2010.

[2]   Noncovalent interactions in the synthesis and design of new compounds. A.M. Maharramov, K.T. Mahmudov, M.N. Kopylovich, A.J.L. Pombeiro, (Eds.), John Wiley & Sons, Inc., Hoboken, NJ, 2016.
[3]   Noncovalent Forces; S. Scheiner, Ed.; Springer, Dordrecht, 2015.

[4]    Noncovalent Interactions in Catalysis, K.T. Mahmudov, M.N. Kopylovich, M.F.C. Guedes da Silva, A.J.L. Pombeiro, (Eds.) Royal Society of Chemistry, UK, 2019.

[5]  Noncovalent Interactions in Proteins, A. Karshikoff, Imperial College Press, 2006.

and reviews

On synthesis
1) Cation−π Interactions in Organic Synthesis
Chem. Rev. 2018, 118, 23, 11353–11432
2) Noncovalent interactions in the synthesis of coordination compounds: Recent advances
Coordination Chemistry Reviews, Volume 345, 2017, Pages 54-72.

On catalysis

1) Supramolecular Catalysis in Metal−Ligand Cluster Hosts,
Chem. Rev. 2015, 115, 3012−3035.

2)     M. Raynal, P. Ballester, A. Vidal-Ferran, P. W. N. M. van Leeuwen, Chem. Soc. Rev. 2014, 43, 1734−1787.

3)   M. Raynal, P. Ballester, A. Vidal-Ferran, P. W. N. M. van Leeuwen, Chem. Soc. Rev. 2014, 43, 1660−1733.
4) Noncovalent interactions in metal complex catalysis, Coordination Chemistry Reviews
Volume 387, 15 May 2019, Pages 32-46.

should be cited in introduction section.

Author Response

Thank you very much for your helpful comments and suggestions. We have added all references that you pointed out.

Reviewer 3 Report

This paper deals with one of the universal weak interaction : the π-π stacking of aromatic species. This is well documented in organic chemistry but less in biochemistry.  In that sense, this review would be interesting and useful for readers.
Alas, I have one regret : This is a compendium of all the works the authors published in the past on cooper/nickel phenoxy complexes(23 papers!..) As a review it should cover a wider subject than cycling through the main author's  publications. I think this paper is not exactly what is called a "review" . 

I only focus on the last sentence of the introduction :

"This review discusses various aspects of  GO and its model complexes including the structural feature of GO, π-π stacking inter-actions of the phenoxyl radical species with indole ring, and comparison of the reactivity without the π-π stacking interaction. 

 Difficult to follow the logic that conduct from GO to the chemical compounds of di-methylthio-phenolate/SbCl6 series and to their indole-linked derivatives. and vice-versa. The discussion about the role of the thio-alkylation in GO is not supported by the "bestiary" of chemical compounds models analysed here that never bear this post-traduction linkage in their structures.

The paper is rigorously and well written I confess, but it  can be published only after major revision by removing all comparisons with GO. The molecules reviewed here have nothing in common with the active site of GO that serve only as "harpooning" the reader. They are flat compounds that cannot escape to inter-molecular  π-stacking, while in GOs the stacking is locally, isolated, without any interaction in the packing with itself.

Author Response

Thank you very much for your helpful comments and suggestions. We have revised all the sentences concerning to comparisons with GO. 

Round 2

Reviewer 3 Report

Going through the re-submission (corrected at minima) it looks better than the first version. May I suggest to change in the title the "of" to "in" ? I let the editor to see if this is better looking. 
With the marks of corrections, it is difficult to follow the reference list but this is ok. I still do not understand why the authors want to absolutely refer  to galactose oxidase in their work. After all, it would be better to refer to DNA stackings that are much closer to the molecules they are talking about.